# Charge Transfer Complex of Lorlatinib with Chloranilic Acid: Characterization and Application to the Development of a Novel 96-Microwell Spectrophotometric Assay with High Throughput

**DOI:** 10.3390/molecules28093852

**Published:** 2023-05-01

**Authors:** Hany W. Darwish, Ibrahim A. Darwish, Awadh M. Ali, Halah S. Almutairi

**Affiliations:** 1Department of Pharmaceutical Chemistry, College of Pharmacy, King Saud University, P.O. Box 2457, Riyadh 11451, Saudi Arabiaalhalah@ksu.edu.sa (H.S.A.); 2Department of Analytical Chemistry, Faculty of Pharmacy, Cairo University, Kasr El-Aini St., Cairo 11562, Egypt

**Keywords:** lorlatinib, non-small cell lung cancer, charge transfer reaction, chloranilic acid, microwell spectrophotometric assay, high-throughput analysis

## Abstract

Lorlatinib (LRL) is the first drug of the third generation of anaplastic lymphoma kinase (ALK) inhibitors used a first-line treatment of non-small cell lung cancer (NSCLC). This study describes, for the first time, the investigations for the formation of a charge transfer complex (CTC) between LRL, as electron donor, with chloranilic acid (CLA), as a π-electron acceptor. The CTC was characterized by ultraviolet (UV)-visible spectrophotometry and computational calculations. The UV-visible spectrophotometry ascertained the formation of the CTC in methanol via formation of a new broad absorption band with maximum absorption peak (λmax) at 530 nm. The molar absorptivity (ε) of the complex was 0.55 × 10^3^ L mol^−1^ cm^−1^ and its band gap energy was 2.3465 eV. The stoichiometric ratio of LRL/CLA was found to be 1:2. The association constant of the complex was 0.40 × 10^3^ L mol^−1^, and its standard free energy was −0.15 × 10^2^ J mole^−1^. The computational calculation for the atomic charges of an energy minimized LRL molecule was conducted, the sites of interaction on the LRL molecule were assigned, and the mechanism of the reaction was postulated. The reaction was adopted as a basis for developing a novel 96-microwell spectrophotometric method (MW-SPA) for LRL. The assay limits of detection and quantitation were 2.1 and 6.5 µg/well, respectively. The assay was validated, and all validation parameters were acceptable. The assay was implemented successfully with great precision and accuracy to the determination of LRL in its bulk form and pharmaceutical formulation (tablets). This assay is simple, economic, and more importantly has a high-throughput property. Therefore, the assay can be valuable for routine in quality control laboratories for analysis of LRL’s bulk form and pharmaceutical tablets.

## 1. Introduction

The term “charge transfer complex” (CTC) was initially conceived of by Mulliken [1], and it was later popularized by Foster [2], who wrote extensively on the topic. During the formation of CTCs, a portion of the electronic charge that is carried by one molecule is passed on to another, which accepts the charge carried by the electron donor molecule [1,2,3]. The chemistry of CTC formation is essential to different fields of the chemical and biochemical sciences, as well as to the processes involved in the bio-electrochemical transfer of energy, biological systems, and pharmaceutical manufacture. Investigating the CTCs has enabled the thorough characterization of a variety of biological processes, such as the possible binding of drugs to receptors, the catalysis of enzymes, and the transport of substances through lipophilic membranes located in various bodily compartments [4,5,6,7,8,9]. The production of CTCs is also essential in another fields, such as the manufacture of materials that are optically and electrically conductive [10,11,12,13], surface chemistry [14], photocatalysts [15], and others [16,17,18]. Additionally, the formation of CTCs between drugs and specific σ- and π-acceptors has been successfully utilized as the basis for the design of high-accurate assays for both qualitative and quantitative detection of pharmaceuticals in raw material and/or dosage forms [19,20,21,22,23,24,25,26,27,28,29,30]. This has been accomplished by effectively employing CTC generation as the basis for the design of high-accurate assays. Over the course of the past few years, we have been conducting research into the development and usage of CTCs in a wide range of pharmaceuticals that contain a variety of acceptors including kinase inhibitors, antibiotics, and others [19,20,21,22,23,24,25,26,27,28,29,30]. The primary objective of our study was to gain a grasp of the chemistry underlying the CT reactions of these medications, namely. how their electrons are donated to a range of electron acceptors. Both the characteristics of CT complexes and the circumstances that lead to their development have been subjected to in-depth investigation.

Lorlatinib (LRL) is the first drug that belongs to the third generation of anaplastic lymphoma kinase (ALK) inhibitors. The chemical structure of LRL is given in Figure 1A, and its name is (10R)-7-amino-12-fluoro-10,15,16,17-tetrahydro-2,10,16-trimethyl-15-oxo-2H-4,8-methenopyrazolo [4,3-h] [2,5,11] benzoxadiaza-cyclotetradecine-3-carbonitrile. It is an orally available ATP-competitive selective inhibitor of ALK as well as the related C-ros oncogene 1 (ROS1) [31]. ALK has a crucial function in nervous system development, and its dysfunction is associated with a variety of malignancies; ROS1 plays a significant role in cancer cell growth and survival. It is now widely accepted that ALK inhibitors are effective drugs in the treatment of ALK-positive non-small cell lung cancer (NSCLC) [32]. Nevertheless, with time, their efficiency tends to decay due to the appearance of mutations that are insensitive to treatment with first- and second-generation ALK inhibitors [31,32]. Consequently, the emergence of the third-generation ALK inhibitors, led by LOR, provided a special interest for this group in treating NSCLC patients suffering from the resistance to the old ALK inhibitors. LOR was firstly approved by the US-FDA in 2018. Subsequently, it has been approved by the EMA in 2019 for the treatment of patients with previously treated advanced ALK-positive NSCLC. Recently, in 2022, it was subjected to an expanded approval in that it was considered a first-line treatment option in advanced ALK-positive NSCLC [33,34]. LRL is available in markets as conventional tablets, with the trade name of Lorbrena^®^ tablets [33].

The therapeutic efficacy of LRL and the presence of potential electron-donating sites on its structure (amine nitrogen and ether oxygen atoms) prompted our interest in gaining a deeper knowledge of the chemical nature of LRL’s CT interaction. In our earlier research, which utilized a wide range of polyhalo- and polycyanoquinone electron acceptors [35], we found that chloranilic acid (CLA, Figure 1A) was the electron acceptor with the highest degree of reactivity. In most circumstances, the CLA reaction takes place instantly at room temperature. In this work, LRL was first reacted with CLA in methanol. The reaction conditions were then optimized, and the association constant of the complex as well as the molar ratio of the reaction were computed. After that, the chemical reaction mechanism was postulated on the basis of the computational calculations. In end, the CT reaction was utilized to develop a new high-throughput 96-microwell spectrophotometric assay (MW-SPA) for the determination of LRL in bulk and its dosage forms (Lorbrena^®^ tablets).

## 2. Results and Discussion

### 2.1. UV-Visible Absorption Spectra and Band Gap Energy

In the range of 200–800 nm, the UV-visible absorption spectra of LRL (0.49 × 10^−3^ M) and CLA (4.8 × 10^−5^ M) solutions were measured. LRL exhibits two discrete absorption maxima (λ_max_) at 300 and 388 nm, but no UV absorption after 440 nm, as evidenced by its spectrum (Figure 1). CLA has a high UV absorption prior to 300 nm and a low UV absorption between 300 and 440 nm (Figure 1B). Various quantities of LRL solution (3.1 × 10^−4^ to 1.2 × 10^−3^ M) were mixed individually with a fixed concentration of CLA solution (5.98 × 10^−3^ M) at 25 ± 2 °C. The absorption spectra of the reaction mixtures were examined relative to the CLA reagent blank solution (Figure 1B), where a violet-colored product with maximum absorption at 530 nm was seen, and its intensity increased with the increasing LRL concentration (Figure 1B).

This behavior supports the hypothesis that a new reaction product will arise. The resulting absorption band had the same pattern and shape as the radical anion of CLA, as reported in the literature [26,35]. Therefore, it was postulated that the reaction was a CT interaction between LRL as an electron donor (D) and CLA as a π-electron acceptor (A). This reaction was carried out in a polar solvent such as methanol to produce CTC (D-A), which was then dissociated by the powerful ionizing ability of methanol to produce the radical anion of CLA:
$$D+A⇌\underset{\text{Complex}}{\left(D\text{-}A\right)}\overset{\text{Polar}\;\text{solvent}}{⇌}\underset{\text{Radical}\;\text{ions}}{D^{\cdot^{+}}+A^{\cdot^{-}}}$$

It is generally known that CLA has two stable forms: a pale violet form(A^2−^) that is stable at higher pH levels, and a purple form (HA^−^) that is stable at pH = 3 [36]. The purple reaction product that was produced when LRL and CLA interacted in methanol was therefore identified as the HA^−^ form of CLA that was involved in the reaction.

The disappearance of the purple color of the reaction mixture upon the addition of mineral acids provided more evidence that the reaction was CT.

We performed a calculation to determine the band gap energy (Eg), which was previously described as “the least energy required for the excitation of an electron from the lower energy valence band into the higher energy for involvement in building a conduction band” [37]. In order to create a Tauc plot and execute Eg calculations, an absorption spectrum of the CTC complex was employed (Figure 2). Plotting energy values (*h*, in eV) vs. (*h*)^2^ results in the Tauc graphic. We were able to determine the value of Eg by projecting the graph’s linear section to the point where (*hυ*)^2^ = 0 [38]. The calculated value of Eg was discovered to be 2.347 eV, per the findings. This low predicted value illustrates how easily electrons go from LRL to CLA and how the CTC absorption band is formed.

### 2.2. Optimization of the Reaction Conditions

The reaction of LRL with CLA was carried out in a number of different solvents, each of which had a unique dielectric constant [39] and polarity index [40]. This was performed in order to establish which solvent was the most suitable for achieving the best possible reaction and color development. In order to accomplish this goal, the absorption spectra of the color that was produced were evaluated and recorded. On the basis of the spectra that were collected, the molar absorptivity (ε) and λ_max_ were determined for each examined solvent. There were some little shifts in the values of max, and there were some changes in the values. The reaction yielded better results in solvents with high polarity and high dielectric constants (such as methanol and ethanol), as opposed to the results obtained in solvents with low polarity and low dielectric constants (e.g., dichloroethane, dioxane), Figure 3. This action could have been the result of a full electron transfer from a molecule of LRL, which acts as an electron donor, to a molecule of CLA, which acts as an electron acceptor. This transfer is favored in polar solvents. In each of the following steps, methanol was chosen to serve as the solvent of choice. The influences of the CLA concentration and time on the reaction were studied, and the results (Figure 4) showed that the optimum CLA concentration was 2% (*w*/*v*) and the reaction occurred instantly in methanol at room temperature (25 ± 2 °C).

### 2.3. Electronic Constants and Properties

#### 2.3.1. Association Constant, Free Energy Change, and Donor Ionization Potential

The Benesi–Hildebrand method [41] was utilized to compute the association constant (K) at room temperature (25 ± 2 °C) and at the λ_max_ of the generated LRL-CLA complex by evaluating the absorption spectra of the complex created by reacting various concentrations of CLA with a constant concentration of LRL. From the obtained straight line (Figure 5), the CTC’s association constant was computed. The association constant value was determined to be 0.4 × 10^3^ L mol^−1^ (Table 1).

The CTC’s standard free energy change (ΔG_0_) is proportional to its formation constant and may be calculated using the formula:

ΔG_0_ = −2.303 RT log Kc

where ΔG_0_ denotes the standard free energy change of the complex (kilojoules; KJ mol^−1^), R donates the gas constant (8.314 KJ mole^−1^), T denotes the absolute temperature in Kelvin (°C + 273), and Kc denotes (as mentioned before) the complex association constant (L mol^−1^). The ΔG_0_ value was −0.12 × 10^2^ J mol^−1^. This value may designate that simplicity and stability of the formed CTC between LRL and CLA [42].

The donor’s (LRL) ionization potential (I_p_) in the CTC of LRL with CLA was computed utilizing Aloisi and Piganatro’s empirical equation [43]:

I_p_ (eV) = 5.76 + 1.53 × 10^−4^ υ_CTC_

where υ_CTC_ denotes the wave number in cm^−1^ of the CTC, and it was found to be 0.92 × 10^2^ eV.

#### 2.3.2. Oscillator Strength and Transition Dipole Moment

The oscillator strength, denoted by the dimensionless number *f*, is the value that is utilized in the description of the transition probability of the CT-band [44]. Using the following formula, oscillator strength (*f*), was derived from the absorption spectrum of the CTC:*f* = 4.32 × 10^−4^ ∫ ε_CTC_^dν^½

where ∫ ε_CTC_^dν^½ denotes the area under the curve of the extinction coefficient of the absorption band of CTC vs. frequency to a first approximation:
*f* = 4.32 × 10^−4^ ε_CTC_^Δν^½
where ε_CTC_ denotes the maximum extinction coefficient of the band and ^Δν^½ is the half-width (i.e., the width of the band at half the maximum extinction). The calculated *f* of the formed CTC was found to be equal 0.4331.

The extinction coefficient is related to the transition dipole moment (µ) of the CTC. It was computed by applying the Tsubumora and Lang [45] equation:

µ = 0.0958 [ε_CTC_ × Δν½/Δν]^½^

where Δν ≈ ν ε_CTC_ is the wave number unit. The value of transition dipole moment = 0.12 × 10^2^ Debye.

#### 2.3.3. Resonance Energy

The method developed by Briegleb and Czekalla [46] was applied in order to calculate the complex’s resonance energy, denoted by the symbol RN:

R_N_ = 7.7 × 10^−4^/[(h ν_CTC_/R_N_) − 3.5]
where ν_CTC_ represents the peak frequency of the CTC, and R_N_ is the resonance energy of the CTC in its ground state. It should be noted that R_N_ is one of the factors that contributes to the stability constant of the complex (a ground state property). It was calculated that the value of R_N_ for the CTC was equal to 0.6688 Debye. Table 1 provides a concise description of the constants and electrical characteristics of the CTC of LRL with CLA.

### 2.4. Molar Ratio and Computational Charge Calculation

The molar ratio of LRL to CLA was calculated, and it was found to be 1:2 (Figure 6), indicating that two electron-donating sites on the LRL molecule contributed to the formation of the CTC with CLA. On the energy-minimized LRL molecule (Figure 7), electron density on each atom was directly computed in order to assign this site among the numerous electron-donating sites available on the LRL molecule. The outcomes are shown in Table 2. The nitrogen atoms with the numbers N10 (aromatic nitrogen in a five-membered ring) and N25 (aniline nitrogen) had the highest electron densities. The relative electron densities for them were −0.7068 and −0.9, respectively; the negative signs imply negative electron densities. These findings suggested that these two nitrogen atoms were involved in the formation of the CTC of LRL with CLA. Accordingly, considering the molar ratio, the reaction mechanism was suggested as described in Figure 8.

### 2.5. Development of MW-SPA

#### 2.5.1. Strategy and Design of the Assay

As of right now, LRL is one of the most effective, powerful, and selective ALK tyrosine kinase inhibitors available for the treatment of ALK-positive NSCLC. Therefore, the efficacy of this drug, as well as the safety of its usage, are dependent on the quality of its dosage form (Lorbrena^®^ tablets). The use of high-performance liquid chromatography (HPLC) with ultraviolet (UV) detection [47] and high-performance liquid chromatography with tandem mass spectrometric detection [48,49,50,51,52] are the analytical techniques that have been published for quantifying LRL. The standards for analytical methods for quality control of pharmaceutical compounds require other analytical approaches rather than those chromatographic ones as they rely on complicated procedures and a demanding apparatus that are very expensive. In addition, each of these methods was created for the purpose of determining the LRL in biological samples, with the exception of the HPLC with UV detection method [47], which was designed for the purpose of quantifying LRL in its dosage forms.

Photometric techniques are essential in pharmaceutical analysis because they may be easily automated with photometric analyzers, which are frequently employed for serial examination of pharmaceutical formulations [53,54]. According to a comprehensive literature assessment, there is no photometric method for determining LRL content in its bulk and/or capsules. Therefore, the purpose of the present study is to develop a photometric method for quantifying LRL in its tablets. Clearly, the chemical structure of LRL contains chromophoric moieties (Figure 7); thus, it is anticipated to exhibit a high molar absorption coefficient. Consequently, the development of a photometric approach for LRL quantification on the basis of its native ultraviolet (UV) light absorption is probable; nonetheless, the availability of a visible photometric method for LRL quantitation is essential for its automation with colorimetric analyzers. In order to construct a photometric methodology for LRL, the aforementioned CT reaction of LRL with CLA was used as a cornerstone. The photometric procedures that make use of colored CTC production and the existing manual technique have only a low throughput [55,56], and in addition to this, they use up a significant amount of organic solvent. As a consequence of this, these tests are expensive, and in addition, they put the analyst in danger by exposing them to the potentially hazardous effects of organic solvents [57,58]. Using an absorbance plate reader, Darwish et al. [21,22,23,24,25,26,27,28,29,30] produced a number of successful microwell-based photometric approaches. These methods were used to estimate the concentration of active substance contained in pharmaceutical dosage forms. These techniques have a high throughput and use only a minimal amount of the organic solvent. As a direct result of this, the emphasis of the present work was placed on the development of a comparable technique for LRL.

#### 2.5.2. Optimization of MW-SPA Conditions

In order to optimize the performance of the reaction in the 96-well assay plate, the experimental conditions were optimized. A summary of these conditions along with the optimum values are given in Table 3.

### 2.6. Validation of MW-SPA

#### 2.6.1. Linear Range and Sensitivity

The dataset was subjected to linear regression, and a calibration curve was constructed using the least-squares approach under MW-SPA-adjusted conditions. As shown in Figure 9, it was discovered that the curve was linear with a strong correlation coefficient in the range of 5–200 µg/well. The intercept, slope, and correlation coefficient of a specific linear fit are shown in Table 4 for clarity. The limit of detection (LOD) and limit of quantitation (LOQ) were estimated in accordance with the International Council for Harmonization (ICH) criteria [59]. The LOD and LOQ values were estimated to be 2.1 and 6.5 µg/well, respectively. A list of the calibration and validation parameters for the constructed MW-SPA can be found in Table 4.

#### 2.6.2. Precision and Accuracy

The precisions of the proposed MW-SPA were evaluated by analyzing samples of LRL solution with varied concentration levels. The findings are reported in Table 5. Precision was measured using relative standard deviations (RSD), which ranged from 1.42 to 2.15% for intra- and 1.80 to 2.45% for inter-assay precisions. These relatively low RSD readings served as reassurance of the assay’s high degree of precision. For the purpose of evaluating the assay’s accuracy, recovery studies were carried out using the identical LRL concentration levels, as were used in the precision studies. The fact that the recovery values ranged from 98.2 to 100.2 percent, as shown in Table 5, indicates that the proposed assay is remarkably accurate.

#### 2.6.3. Specificity and Interference

In order to avoid any potential interference from UV-absorbing inactive ingredients, which may be extracted from the pharmaceutical formulation of LRL (Lorbrena^®^ tablets), measurements in the proposed MW-SPA are carried out in the visible range. This is done so that any potential interference can be avoided. Because the assay described in this study included extracting LRL from Lorbrena^®^ tablets using methanol, the excipients and inactive components did not dissolve, which demonstrated the assay’s high level of specificity.

### 2.7. Application of MW-SPA to the Quantitation of LRL in Lorbrena^®^ Tablets

The advantageous usage of the suggested MW-SPA to assess LRL in Lorbrena^®^ tablets was made possible as a result of the impactful validation results. The obtained labeled amount had a mean value of 99.1% with a standard deviation of 0.65 (Table 6). This result exhibited that the suggested MW-SPA is appropriate for determining the LRL concentration in Lorbrena^®^ tablets.

## 3. Materials and Methods

### 3.1. Apparatus

All the characteristics of the apparatuses adopted in this work including the UV-VIS spectrophotometer, absorbance microplate reader, 96-microwell assay plates, and 8-channel pipette were mentioned in our previous report [35].

### 3.2. Chemicals and Materials

Selleck Chemicals LLC (Houston, TX, USA) was the vendor for the acquisition of LRL. Lorbrena^®^ tablets (manufactured by Pfizer Inc., New York, NY, USA) were kindly donated by the Saudi Food and Drug Authority (SFDA: Riyadh, Saudi Arabia) labelled to contain 100 mg of LOR per tablet. Sigma-Aldrich Corporation was the vendor for CLA (St. Louis, MO, USA). In the course of the research, analytical-grade solvents and reagents were utilized (Fisher Scientific, California, CA, USA).

### 3.3. Preparation of LRL Standard and Tablet Solutions

A quantity of LRL (100 mg) was transferred to a 50 mL volumetric flask, and 20 mL of methanol was added. The volume was completed with methanol to yield a final concentration of 2 mg mL^−1^. For the preparation of the tablets solution, 10 tablets were crushed and finely powdered. A specified weight equivalent to 100 mg LRL was transferred to a 50 mL volumetric flask. The flask was subjected to sonication for a period of thirty minutes, after which it was diluted with the same solvent up to the mark. After passing through chromafil^®^ Xtra 0.2 m filter paper, this solution that contained 2 mg mL^−1^ of LRL was diluted with methanol in order to achieve the required concentrations of LRL for the subsequent analysis by the procedures of the proposed MW-SPA.

### 3.4. Calculation of Association Constant and Molar Ratio

We calculated the association constant of the CTC with the use of the Benesi–Hildebrand method [41]. The spectrophotometric titration [60] and Job’s continuous variation [61] methods were used for calculating the molar ratio of LRL to CLA.

### 3.5. Calculation of Electron Densities on Atoms

The electron densities on LRL atoms were calculated utilizing the CS Chem3D Ultra, (Cambridge Soft Corporation, Cambridge, MA, USA), equipped with molecular orbital computations software (MOPAC, version 16.0) and molecular dynamics computations software, wherein the energy minimization and charge calculation were carried out.

### 3.6. MW-SPA Procedures

Aliquots (100 µL) of the standard or tablet sample solution that includes varying concentrations of LRL (5–200 µg/well) were transferred into each well of the transparent 96-microwell assay plates. After adding one hundred microliters of a CLA solution containing 2% (*w*/*v*), the reaction was instantaneously proceeded at room temperature (25 ± 2 °C). At a wavelength of 490 nm, the absorbances of the solutions were measured with the use of the absorbance microplate reader.

## 4. Conclusions

The results of a UV-visible investigation showed that LRL and CLA combined to produce CTC. This was evidenced by the emergence of a new distinct absorption band at 530 nm, which occurred during the reaction between the two substances in methanol. It was discovered that a ratio of 1:2 exists inside the complex’s molecular stoichiometry. Both the polarity index and the dielectric constant of the solvent that was used for the reaction had an effect on the values of the λ_max_ and ε of the formed complex. The computational charge calculation and the molecular modeling were able to point to the potential location of the two interaction sites on the LRL molecules that played a role in the development of the complex. In this study, the interaction between LRL and CLA was created to serve as the basis for a novel 96-microwell spectrophotometric assay for the estimation of LRL in both bulk powder and tablets. The microwell-based spectrophotometric test for LRL that was developed in this paper is regarded to be the first assay of its kind for LRL determination. This assay is distinguished by its high throughput, which makes it possible to conduct the analysis of a substantial number of samples in a very short amount of time. Because it only needs a small quantity of organic solvent to be used, it also has the benefit of being friendly to the environment when it is employed in pharmaceutical quality control laboratories.

## Figures and Tables

**Figure 1 molecules-28-03852-f001:**
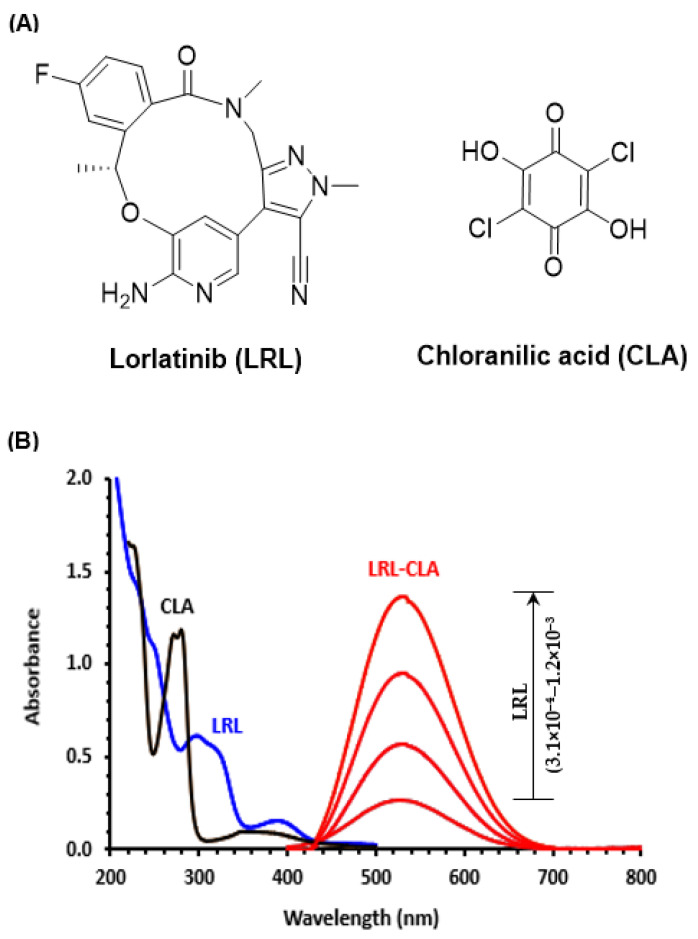
(**A**) The chemical structures and abbreviation of lorlatinib and chloranilic acid. (**B**) Absorption spectra of LRL (0.49 × 10^−3^ M), CLA (4.8 × 10^−5^ M), and reaction mixtures (LRL-CLA) containing varying concentrations of LRL (3.1 × 10^−4^ M-1.2 × 10^−3^ M) and a fixed concentration of CLA (5.98 × 10^−3^ M); all solutions were in methanol.

**Figure 2 molecules-28-03852-f002:**
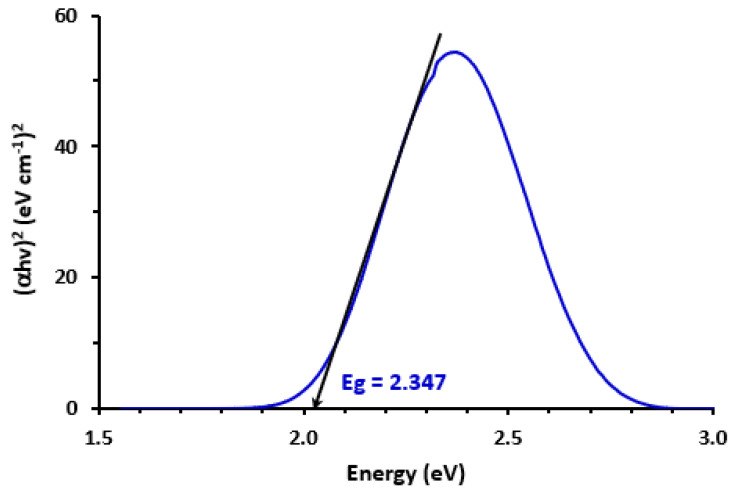
Tauc plot of energy (*hυ*) against (α*hυ*)^2^ for the CTC of LRL with CLA in methanol.

**Figure 3 molecules-28-03852-f003:**
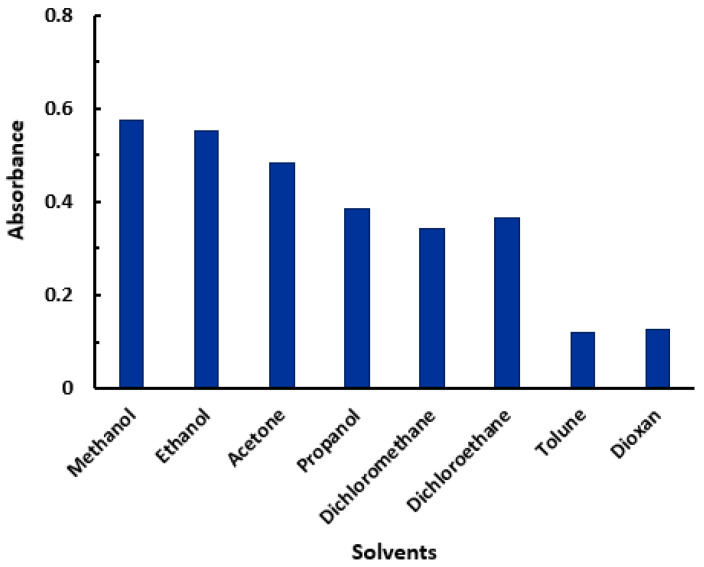
The effect of solvent on the CT reaction between LRL and CLA. The concentrations of LRL and CLA in the reaction were 6.73 × 10^−4^ and 3.1 × 10^−4^ M, respectively. The absorbances of the reaction mixture were measured at 530 nm.

**Figure 4 molecules-28-03852-f004:**
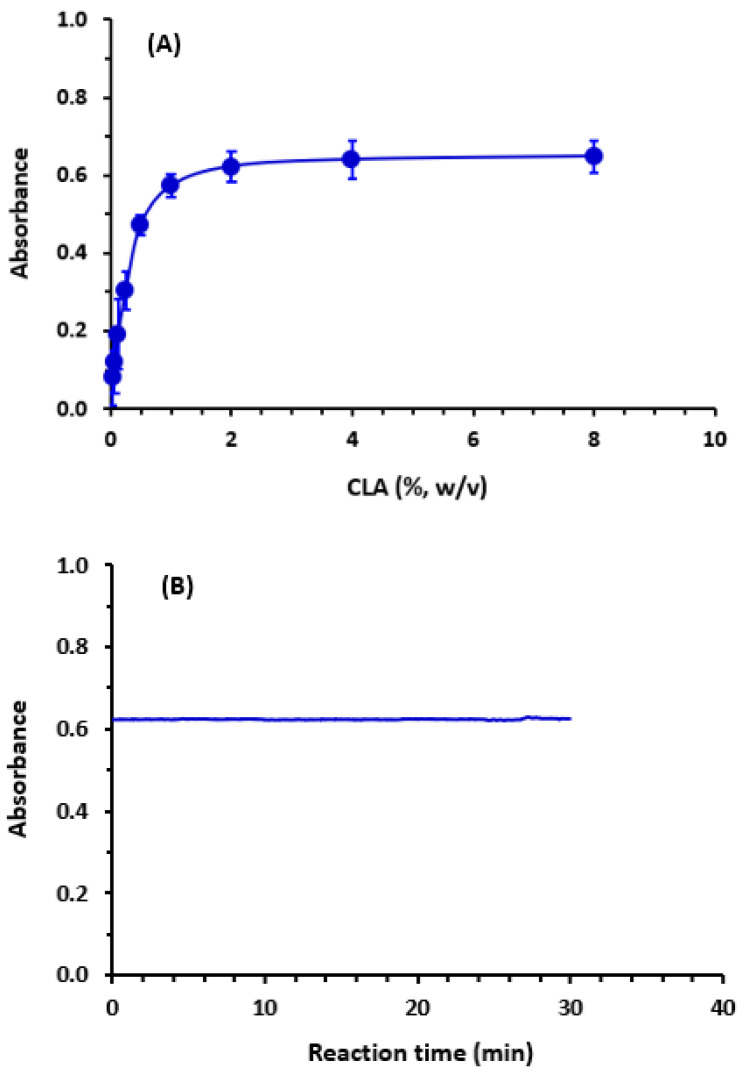
The effect of CLA concentration (**A**) and the kinetic profile of the CT reaction between LRL and CLA (**B**). The concentrations of LRL and CLA in the reaction were 6.73 × 10^−4^ and 3.1 × 10^−4^ M, respectively. The absorbances of the reaction mixture were measured at 530 nm. Absorbances in (**A**) are the mean of 3 measurements ± SD.

**Figure 5 molecules-28-03852-f005:**
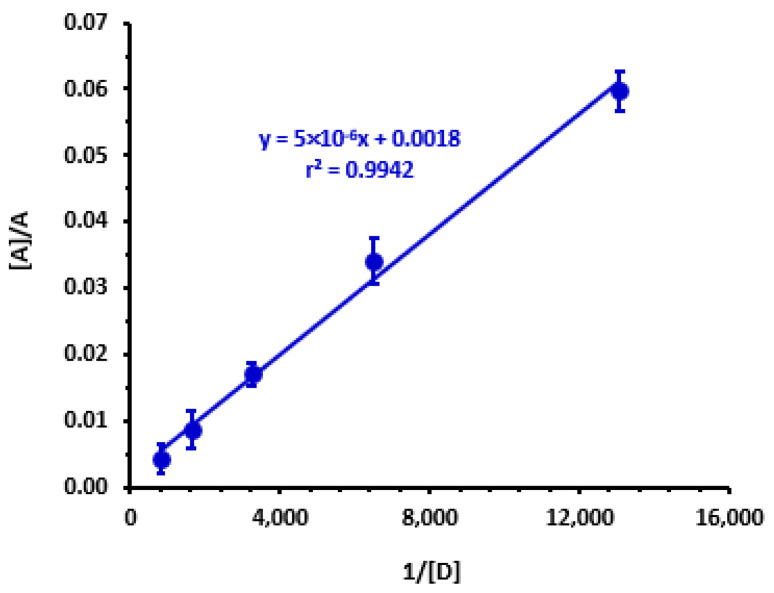
Benesi–Hildebrand plot for formation of the CTC of LRL with CLA. Linear fitting equation and correlation coefficient (*r*) are provided on the plot. [A], A, and [D] are the molar concentrations of CLA, absorbances of the complex reaction mixture, and molar concentration of LRL, respectively. Values are the mean of 3 measurements ± SD.

**Figure 6 molecules-28-03852-f006:**
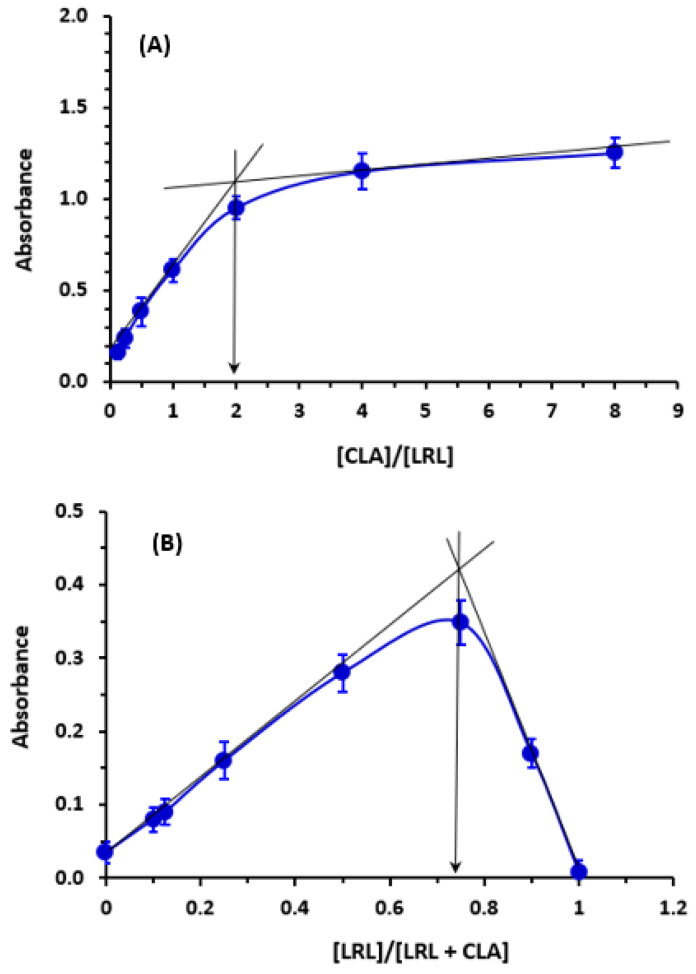
The spectrophotometric titration (**A**) and Job’s continuous variation (**B**) plots for determination of molar ratio of the CT reaction of LRL with CLA. Values are the mean of 3 measurements ± SD.

**Figure 7 molecules-28-03852-f007:**
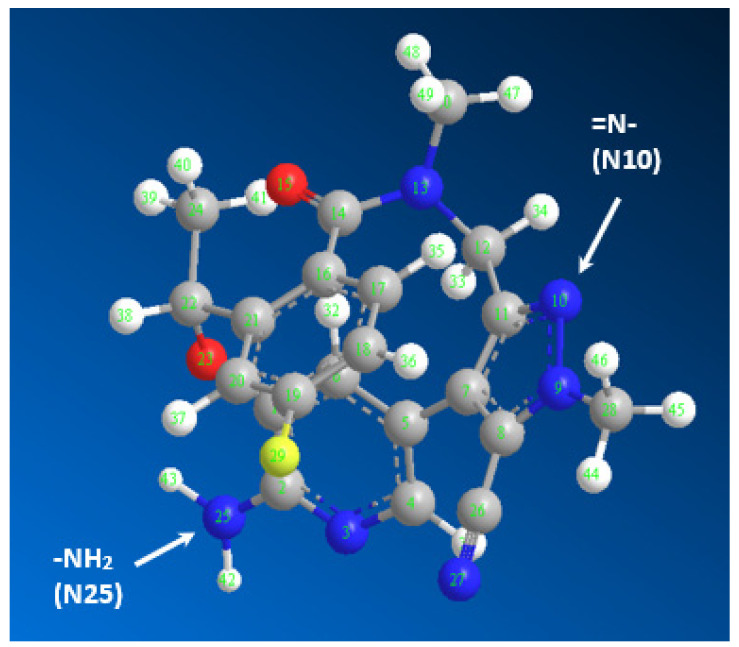
The energy minimized LRL molecule with atom numbers, and the arrows point to the atoms having the highest electron densities (N10 and N25).

**Figure 8 molecules-28-03852-f008:**
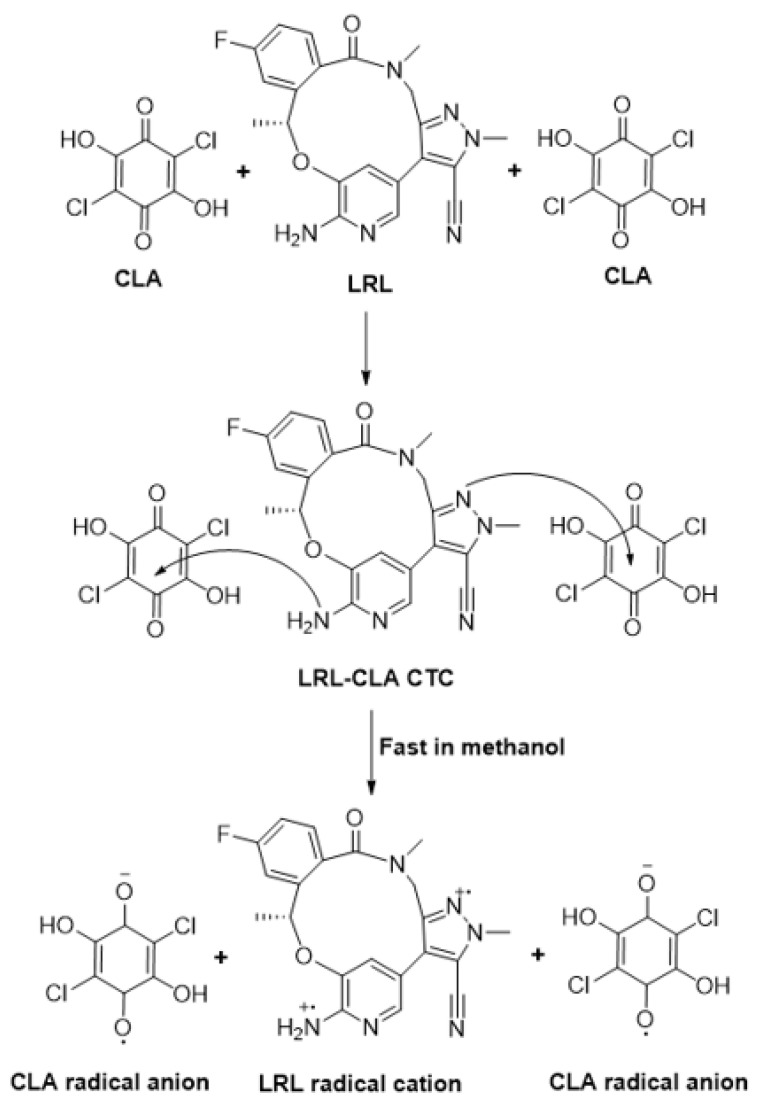
The scheme for the formation of the CTC between LRL and CLA.

**Figure 9 molecules-28-03852-f009:**
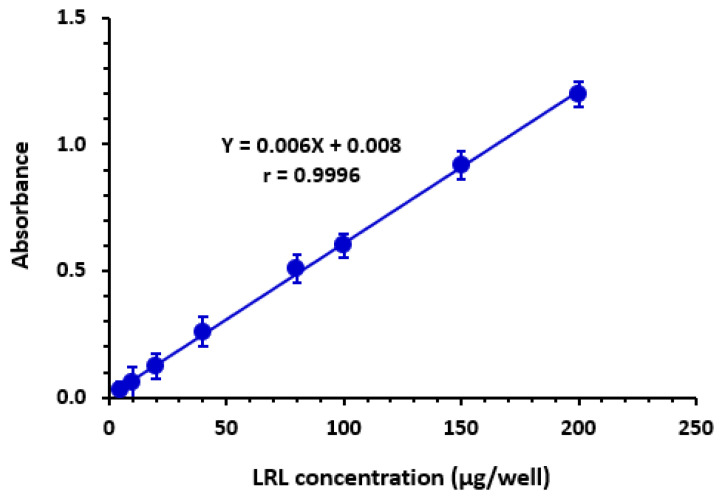
The calibration curve for determination of LRL by the proposed MW-SPA via formation of CTC with CLA. The linear regression equation and its correlation coefficient (*r*) are provided on the calibration line. The experimental conditions: CLA concentration was 2% (*w*/*v*), solvent was methanol, reaction time was 5 min, and the measurements were carried out at 530 nm.

**Table 1 molecules-28-03852-t001:** Electronic constants/properties of the CTC of LRL with CLA.

Constant/Property	Value
Molar absorptivity, ε (L mol^−1^ cm^−1^)	0.55 × 10^3^
Association constant, *K* (L mol^−1^)	0.40 × 10^3^
Ionization potential, I_p_ (eV)	0.92 × 10^2^
Energy, hν (eV)	2.3465
Resonance energy, R_N_ (eV)	0.6688
Dissociation energy, W (eV)	3.2523
Oscillator strength, *f*	0.4331
Standard free energy change, △G^0^ (J mol^−1^)	−0.15 × 10^2^
Transition dipole moment, µ_(Debye)_	0.12 × 10^2^

**Table 2 molecules-28-03852-t002:** Atom numbers, types, and their calculated charges on the energy minimized LRL molecule.

Atom Number	Atom Type	Charge	Atom Number	Atom Type	Charge
C(1)	Aromatic C, in benzene	0.0825	C(20)	Aromatic C, in benzene	−0.15
C(2)	Aromatic C, in benzene	0.41	C(21)	Aromatic C, in benzene	−0.1435
N(3)	Aromatic N, with s lone pair	−0.62	C(22)	Alkyl C, SP3	0.4235
C(4)	Aromatic C, in benzene	0.16	O(23)	Alcohol, ether O	−0.3625
C(5)	Aromatic C, in benzene	0	C(24)	Alkyl C, SP3	0
C(6)	Aromatic C, in benzene	−0.15	N(25)	Enamine, aniline N	−0.9
C(7)	Aromatic C, in 5-ring	0	C(26)	Cyano C	0.5371
C(8)	Aromatic C, in 5-ring	−0.1316	N(27)	Triple bond N	−0.5571
N(9)	Aromatic N, in 5-ring	0.314	C(28)	Alkyl carbon, SP3	0.2556
N(10)	Aromatic N, in 5-ring	−0.7068	F(29)	Fluorine	−0.19
C(11)	Aromatic C, in 5-ring	0.1078	C(30)	Alkyl carbon, SP3	0.3001
C(12)	Alkyl C, SP3	0.4811	H(31)–H(32)	H attached to C	0.15
N(13	Amide N	−0.6602	H(33)–H(34)	H attached to C	0
C(14)	Amide carbonyl C	0.5438	H(35)–H(37)	H attached to C	0.15
O(15)	Carbonyl O, in amide	−0.57	H(38)–H(41)	H attached to C	0
C(16)	Aromatic C, in benzene	0.0862	H(42)–H(43)	H of enamine N	0.4
C(17)–C(18)	Aromatic C, in benzene	−0.15	H(44)–H(49)	H attached to C	0
C(19)	Aromatic C, in benzene	0.19			

**Table 3 molecules-28-03852-t003:** Optimization of experimental conditions of MW-SPA for LRL via formation of colored CTC with CLA.

Condition	Studied Range	Optimum Value
CLA conc. (%, *w*/*v*)	0.1–8	2
Solvent	Different ^a^	Methanol
Reaction time (min)	0–30	Instantaneous ^b^
λ_max_ (nm)	400–800	530 ^c^

^a^ Solvents used are provided in Figure 3. ^b^ Measurements on the plate reader were carried out within 5 min. ^c^ Measurements on the plate reader were carried out at 530 nm.

**Table 4 molecules-28-03852-t004:** Calibration parameters for the determination of LRL by the proposed MW-SPA on the basis of the formation of colored CTC with CLA.

Parameter	Value
Linear range (µg/well)	5–200
Intercept (a)	0.008
Standard deviation of intercept (SDa)	3.9 × 10^−3^
Slope (b)	0.006
Standard deviation of slope (SDb)	2.8 × 10^−3^
Correlation coefficient (*r*)	0.9996
Limit of detection (LOD, µg/well)	2.1
Limit of quantitation (LOQ, µg/well)	6.5

**Table 5 molecules-28-03852-t005:** Precision of the proposed MW-SPA at different LRL concentration levels.

Taken Concentration (µg/well)	Precision: Relative Standard Deviation (%)		Accuracy: Recovery (% ± SD) ^a^
	Intra-Assay, *n* = 3	Inter-Assay, *n* = 6	
12.5	2.15	2.45	99.6 ± 2.3
25	1.81	2.36	98.2 ± 1.8
50	1.53	2.14	99.1 ± 1.2
100	1.62	1.80	100.2 ± 2.4
200	1.42	1.82	99.1 ± 2.2

^a^ Values are mean of three determinations.

**Table 6 molecules-28-03852-t006:** Analysis of Lorbrena^®^ tablets by the proposed MW-SPA.

Nominal Concentration (μg/mL)	Found Concentration ^a^ (μg/mL)		Recovery ^a^ (%)
125	122.75		98.2 ± 2.3
500	497		99.4 ± 1.2
1000	991		99.1 ± 2.4
2000	1994		99.7 ± 2.2
		Mean	99.1
		SD	0.65

^a^ Values are mean of three determinations.

## Data Availability

All data are in the article.

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
