# Peer review of "Charge Transfer Complex of Lorlatinib with Chloranilic Acid: Characterization and Application to the Development of a Novel 96-Microwell Spectrophotometric Assay with High Throughput"

_molecules, 2023, doi:10.3390/molecules28093852_

Round 1

Reviewer 1 Report

The authors missed all figures and tables in the manuscript. Please add all illustrations in the text.

The authors missed all figures and tables in the manuscript. Please add all illustrations in the text.

Author Response

Please, find the attached file.

Reviewer 2 Report

The research paper »Charge transfer complex of lorlatinib with chloranilic acid: characterization and appli-cation to development of novel 96-microwell spectrophotometric assay with high throughput« of H.W. Darwish et al is about the study of formation of charge transfer complex between lorlatinib (a molecule used as an inhibitor in cell lung cancer curing) and chloranilic acid. The study is supplemented with a development of 96-microwell spectrophotometric assay for quantitative determination of lorlatinib quality control laboratories.

Although the topic might be interesting for readers of Molecules journal and perhaps could even lead to adoption of the proposed assay for routine quantitative analysis of lorlatinib, my suggestion is to reject the manuscript for publication because the manuscript was received in an unfinished form without experimental results being presented.

Namely, the manuscript seems to be sent in a big hurry and as such it cannot be regularly reviewed.

1.)    No Figures and Tables can be found in the manuscript although the authors are referencing to 8 figures and 6 tables that should be included in the manuscript.

2.)    One can find many hyphens in the paper where they should not be present (e.g. one already in the title of the manuscript »appli-cation« [Line 3]) what makes reading the manuscript very difficult. In the manuscript also special characters are missing as well as the text is not formatted properly (in places where the text should be super- or subscripted only spiral symbols can be found (see e.g. lines 17 and 18). I believe that it should be the responsibility of the authors to submit for a review the manuscript in a form that is suitable for a review.

3.)    While perhaps the upper two points may still be attributed to some technical problems, there are also other indices that the manuscript was not prepared in a manner that is suitable for reviewing. See, e.g. unfinished sentences (» There were some little shifts in the values of max, and there were some changes in the values of. « [Lines 135-136]) and claims that are clear nonsense (e.g. »The reaction yielded better results in solvents with high polarity and high dielectric constants (such as methanol and acetonitrile), as opposed to the results obtained in solvents with low polarity and low dielec-tric constants (such as water and ethanol)« [lines 136-138] (water is known to be more polar and has higher dielectric constant than methanol and acetonitrile, while ethanol is less polar and has lower dielectric constant than methanol and acetonitrile) ).

The manuscript is written in readable and understandable English, however, some words could be more appropriately chosen and some sentences rewritten to further facilitate reading of the manuscript.

Author Response

Please, find the attached file.

Round 2

Reviewer 1 Report

Authors did all necessary changes. 

Please check the minor mistakes.

Reviewer 2 Report

The research paper »Charge transfer complex of lorlatinib with chloranilic acid: characterization and appli-cation to development of novel 96-microwell spectrophotometric assay with high throughput« of H.W. Darwish et al is about the study of formation of charge transfer complex between lorlatinib (a molecule used as an inhibitor in cell lung cancer curing) and chloranilic acid. The study is supplemented with a development of 96-microwell spectrophotometric assay for quantitative determination of lorlatinib quality control laboratories.

The paper can be now published as it.

Paper is readable.